# Self-Normalizing Multi-Omics Neural Network for Pan-Cancer Prognostication

**DOI:** 10.3390/ijms26157358

**Published:** 2025-07-30

**Authors:** Asim Waqas, Aakash Tripathi, Sabeen Ahmed, Ashwin Mukund, Hamza Farooq, Joseph O. Johnson, Paul A. Stewart, Mia Naeini, Matthew B. Schabath, Ghulam Rasool

**Affiliations:** 1Department of Cancer Epidemiology, Moffitt Cancer Center and Research Institute, Tampa, FL 33612, USA; matthew.schabath@moffitt.org; 2Department of Machine Learning, Moffitt Cancer Center and Research Institute, Tampa, FL 33612, USA; aakash.tripathi@moffitt.org (A.T.); sabeen.ahmed@moffitt.org (S.A.); ashwin.mukund@moffitt.org (A.M.); ghulam.rasool@moffitt.org (G.R.); 3Department of Electrical Engineering, University of South Florida, Tampa, FL 33620, USA; mnaeini@usf.edu; 4Center for Magnetic Resonance Research, University of Minnesota, Minneapolis, MN 55455, USA; faroo014@umn.edu; 5Analytic Microscopy Core Facility, Moffitt Cancer Center and Research Institute, Tampa, FL 33612, USA; joseph.johnson@moffitt.org; 6Huntsman Cancer Institute, University of Utah, Salt Lake City, UT 84112, USA; paul.stewart@hci.utah.edu; 7Department of Nutrition and Integrative Physiology, University of Utah, Salt Lake City, UT 84112, USA; 8Department of Neuro-Oncology, Moffitt Cancer Center and Research Institute, Tampa, FL 33612, USA; 9Department of Oncologic Sciences, University of South Florida, Tampa, FL 33612, USA

**Keywords:** cancer, oncology, multi-omics, multimodal, pan-cancer, machine learning, deep learning, survival, classification

## Abstract

Prognostic markers such as overall survival (OS) and tertiary lymphoid structure (TLS) ratios, alongside diagnostic signatures like primary cancer-type classification, provide critical information for treatment selection, risk stratification, and longitudinal care planning across the oncology continuum. However, extracting these signals solely from sparse, high-dimensional multi-omics data remains a major challenge due to heterogeneity and frequent missingness in patient profiles. To address this challenge, we present SeNMo, a self-normalizing deep neural network trained on five heterogeneous omics layers—gene expression, DNA methylation, miRNA abundance, somatic mutations, and protein expression—along with the clinical variables, that learns a unified representation robust to missing modalities. Trained on more than 10,000 patient profiles across 32 tumor types from The Cancer Genome Atlas (TCGA), SeNMo provides a baseline that can be readily fine-tuned for diverse downstream tasks. On a held-out TCGA test set, the model achieved a concordance index of 0.758 for OS prediction, while external evaluation yielded 0.73 on the CPTAC lung squamous cell carcinoma cohort and 0.66 on an independent 108-patient Moffitt Cancer Center cohort. Furthermore, on Moffitt’s cohort, baseline SeNMo fine-tuned for TLS ratio prediction aligned with expert annotations (*p* < 0.05) and sharply separated high- versus low-TLS groups, reflecting distinct survival outcomes. Without altering the backbone, a single linear head classified primary cancer type with 99.8% accuracy across the 33 classes. By unifying diagnostic and prognostic predictions in a modality-robust architecture, SeNMo demonstrated strong performance across multiple clinically relevant tasks, including survival estimation, cancer classification, and TLS ratio prediction, highlighting its translational potential for multi-omics oncology applications.

## 1. Introduction

Developing accurate prognostic and diagnostic models is central to advancing precision oncology. Multi-omics data, spanning gene expression, DNA methylation, somatic mutations, protein levels, and miRNA abundance, offer unprecedented opportunities to build such models by capturing diverse molecular signals linked to cancer outcomes and phenotypes [1,2]. However, leveraging these high-dimensional data across tumor types remains a major challenge due to frequent missingness, data heterogeneity, and the lack of scalable integration frameworks. Pan-cancer analysis provides a compelling approach to address these challenges by uncovering both shared and distinct molecular patterns across cancers. However, few existing models can robustly integrate multiple omics modalities and generalize to unseen data or tasks. Addressing this gap requires unified architectures that are modality-robust, clinically relevant, and capable of supporting diverse tasks such as survival prediction, tumor classification, and immune microenvironment assessment.

Prognostic and diagnostic models built on molecular data are an important source for informing treatment selection, stratifying risk, and guiding follow-up strategies in oncology. Overall survival (OS) estimation provides a foundational metric for understanding disease trajectory and evaluating therapeutic impact. Accurate classification of the primary cancer type is equally critical, especially for tumors with ambiguous histology or metastatic presentation, where misdiagnosis can lead to suboptimal care. In parallel, the tumor immune microenvironment, reflected in features like tertiary lymphoid structures (TLS), is gaining recognition as a key determinant of immunotherapy response and long-term outcomes. While these clinical endpoints are often studied in isolation, integrating them into a unified modeling framework can support a more comprehensive view of disease biology and prognosis. A pan-cancer, multi-omics approach offers the potential to power such models by leveraging diverse signals from thousands of patients and tumor types.

Despite promising advances in multi-omics modeling, existing approaches are often constrained by narrow scope, limited modality integration, or weak generalization across tumor types and tasks. Many frameworks are trained on a single omics layer or focus on a specific cancer type, limiting their translational relevance in heterogeneous clinical settings. Others rely on complex feature engineering pipelines that may not scale to large datasets or adapt well to missing data. Deep learning models have shown strong performance in specific applications, but are frequently sensitive to modality dropout, overfitting, and lack of task transferability. Moreover, few models are designed to jointly support survival analysis, diagnostic classification, and immune microenvironment characterization in a single architecture. These gaps highlight the need for a unified, pan-cancer framework that can robustly integrate heterogeneous multi-omics data, support multiple clinically relevant tasks, and generalize across datasets without extensive retraining.

To address these challenges, we developed SeNMo—a self-normalizing deep neural network designed to integrate six heterogeneous data modalities, including gene expression, DNA methylation, miRNA abundance, somatic mutations, protein expression, and clinical variables. Trained on over 10,000 patient samples across 33 tumor types from The Cancer Genome Atlas (TCGA), SeNMo learns a unified, low-dimensional representation that is robust to missing modalities and adaptable to multiple downstream tasks. Without altering the backbone architecture, we fine-tuned SeNMo for three clinically meaningful applications: OS prediction, primary cancer type classification, and TLS ratio estimation. The model’s self-normalizing architecture enhances training stability across high-dimensional inputs, while its generalizability enables deployment on external datasets with minimal adjustment. Together, these features make SeNMo a versatile tool for multimodal cancer prognostication and diagnostic support. Figure 1 presents the entire outline of this work.

This study presents the first unified framework for pan-cancer multi-omics modeling that jointly supports prognostic and diagnostic tasks using a self-normalizing neural architecture. SeNMo demonstrates strong performance in predicting OS, accurately classifying 33 primary cancer types, and estimating TLS ratios from molecular data alone. By incorporating six distinct data modalities and enabling transfer to external cohorts through fine-tuning, SeNMo addresses key limitations of existing models in scalability, robustness, and clinical relevance. Beyond model performance, we contribute the learned patient features (also called embeddings) generated from our model to an open-access latent embedding resource [4], which facilitates further exploration and reuse of the SeNMo’s learned representations. Taken together, our work advances the development of integrative, modality-robust models for multi-omics oncology applications and sets the stage for broader translational use of deep learning in cancer prognosis and diagnosis.

## 2. Results

### 2.1. Prognostic Modeling: Overall Survival (OS)

Pan-cancer performance: Figure 2 summarizes SeNMo’s prognostic ability when trained on the full six-modal TCGA dataset and evaluated on the held-out test set. The inference on the test set showed the C-Index of 0.757, the average of the C-Indices from the 10 checkpoints. To further validate our findings, we created an ensemble of the 10 checkpoints by averaging the prediction vectors from all the models and then evaluating the final averaged prediction vector for C-Index. For the pan-cancer, multi-omics data, SeNMo achieved an ensemble C-Index of 0.758 on the held-out test set. The significance level in all these analyses is 95%, i.e., p<0.05, indicating statistically significant values. Training and validation trajectories are provided in Figure A4.

Impact of missing modalities: Modality ablation experiments confirmed that performance degraded gracefully as data layers were withheld. Using only gene-expression inputs, SeNMo achieved test/ensemble C-indices of 0.718/0.728, respectively. DNA- methylation alone yielded 0.644/0.650, and miRNA alone 0.686/0.702. Combining the three transcriptomic layers recovered much of the full-model performance (0.725/0.726). Adding protein (4-modal) data increased the C-index to 0.746/0.751, while adding mutation (5-modal) data resulted in the C-indices of 0.746/0.749.

Tumor-specific generalization: Applied cancer by cancer on the held-out test set for each of 33 tumor types, the six-modal model achieved statistically significant (p<0.05) C-indices in 29 cancers (Figure 3; Table 1). The best performance occurred in pheochromocytoma and paraganglioma (TCGA-PCPG) with test/ensemble C-indices of 0.900/0.929. Four cohorts (GBM, LAML, PRAD, TGCT) initially fell below significance. Targeted fine-tuning for ten epochs raised the C-indices for GBM, LAML, and PRAD to 0.642/0.650, 0.627/0.626, and 0.541/0.542, respectively, while TGCT was still not statistically significant (i.e., p>0.05) and proved to be a failure case for this study.

External validation: External evaluation demonstrated robustness to domain shift. Zero-shot transfer to CPTAC-LSCC and an independent Moffitt-LSCC cohort produced test/ensemble C-indices of 0.48/0.50 and 0.581/0.590, respectively. A brief ten-epoch fine-tune elevated these scores to 0.677/0.730 (CPTAC-LSCC) and 0.647/0.656 (Moffitt-LSCC), illustrating efficient domain adaptation. These results are highlighted in green box-plots of Figure 3.

Risk stratification: We further investigated the SeNMo’s ability to stratify the patients based on low, intermediate, and high risk conditions. We generate Kaplan–Meier (KM) curves of our model on the pan-cancer, multi-omics held-out test set, as shown in Figure 4. We selected the low/intermediate/high risk terciles stratification as the 33-66-100 percentile of hazard predictions [5,6]. The hazard scores predicted by SeNMo are used to evaluate the model’s stratification ability. The KM comparative analysis shows that SeNMo distinguished the patients across the three groups. The low-risk group (green) exhibited the highest survival probability, maintaining close to 100% survival up to approximately 5 years, and gradually declining to about 60% by the 25-year mark. The intermediate-risk group (blue) showed a significantly lower survival probability, starting to diverge from the low-risk group early on and reaching around 40% by the 15-year mark of the study period. The high-risk group (orange) displayed the most pronounced decline in survival probability, with a steep drop to approximately 20% survival within the first 10 years, and further reducing to below 10% after 10 years. The logrank test to evaluate the significance of this stratification shows that the *p*-value of low vs. intermediate curves is $1.66\times 10^{-5}$, low vs. high is $1.16\times 10^{-46}$, and intermediate vs. high is $1.92\times 10^{-22}$, showing significant results, i.e., p<0.05. The 95% confidence intervals around each curve show the stability of these estimates.

### 2.2. Diagnostic Modeling: Primary Cancer Type Classification

Primary cancer type classification: To test the generalizability of SeNMo to diagnostic tasks, we carried out the prediction of primary cancer type from pan-cancer, multi-omics data. Excluding the *stage* variable from the clinical covariates to avoid label leakage, we framed cancer identification as a 33-way classification task. When considering a cancer type classification problem, the stage adds a bias in the data because of the staging distribution among different cancers. SeNMo achieved near-perfect discrimination with mean training accuracy of 99.9%, validation accuracy of 99.8%, and identical 99.8% accuracy for both single-checkpoint and ensemble inference (Table 2, Figure A5).

Error distribution: The confusion matrix (Figure A5) depicts a clear concentration of values along the diagonal, indicating a high rate of correct predictions across all cancer types. Residual misclassifications are restricted to biologically related pairs (e.g., colon vs rectal adenocarcinoma). The scatter plot shows an alignment of predicted labels with true labels along the diagonal line, highlighting the model’s robust predictive accuracy.

Per-class metrics: The classification report across various cancer types reveals that the model consistently maintains high precision, recall, and F1-scores for all tumor types, reflecting balanced sensitivity and specificity across the highly imbalanced class distribution. Detailed values are provided in Table 2.

Interpretation: The model’s diagnostic power highlights the distinct molecular signatures captured across gene expression, DNA methylation, miRNA abundance, protein levels and mutational spectra. These results demonstrate that SeNMo’s unified representation retains class-specific signals despite training jointly on heterogeneous modalities and objectives.

### 2.3. Immune Microenvironment Modeling: TLS Ratio Prediction

TLS ratio estimation as a window into the immune micro-environment: To test whether SeNMo’s latent representation captures micro-environmental cues, we fine-tuned the backbone on a cohort of lung-squamous cell carcinoma data from Moffitt Cancer Center to predict the tissue-level TLS ratio (segmented TLS area/total tissue area). The task was formulated as a single-output regression problem. The SeNMo baseline model was subjected to few-shots fine-tuning using the cross-validation on 76 patients (80%) and evaluated on the held-out 20-patient (20%) cohort, as shown in Figure 5.

Concordance with expert assessment: On the held-out test set, SeNMo’s continuous TLS predictions were significantly similar to manual annotations (Figure 5b); a paired comparison found no significant mean difference (p=0.064), confirming close agreement.

Discriminating high versus low TLS burden: Violin plots illustrate the separation between manually defined high/low TLS groups and the corresponding SeNMo-predicted groups (Figure 5c). In both cases the distributions differ significantly (p<0.05), indicating that the model preserves biologically meaningful thresholds.

Prognostic relevance of modeled TLS ratios: Kaplan–Meier analysis demonstrates that patients stratified by either manual or SeNMo-predicted TLS ratio exhibit distinct survival outcomes. Manual labels yield a log-rank p=0.019, while SeNMo-based stratification improves the signal to $p=2.5\times 10^{-4}$ (Figure 5d). These findings show that the learned representations not only reproduce pathologist assessments but also retain prognostic power.

## 3. Discussion

SeNMo integrates five omics layers and clinical covariates within a self-normalizing architecture to deliver a unified representation that performs strongly across prognostic, diagnostic, and immune-profiling tasks. We analyzed a pan-cancer dataset of multiple cancer types (with varying amounts of features) using our SeNMo encoder-based framework. Public databases such as CPTAC and TCGA contain common identifiers within their data that connect data from the same patient. Therefore, molecular data, such as gene expression, miRNA expression, DNA methylation, somatic mutations, and protein expression can be consolidated to represent a singular patient. However, such high-dimensional data has intra- and inter-dataset correlations, heterogeneous measurement scales, missing values, technical variations, and other forms of noise [7]. The concordance indices achieved for overall survival (up to 0.758 pan-cancer and 0.730 after domain adaptation) have been shown to outperform the existing works in OS prediction when considering the data modalities included in our data [8]. Moreover, we observed that adding more data and types of modalities increased the model’s performance. Crucially, SeNMo maintains performance when modalities are missing, confirming the utility of self-normalizing layers for handling heterogeneous, sparsely observed inputs. After extensive training-evaluation runs, we found, through optimal parameter searching, a model that performs very well across the different data types and tasks (refer to Figure A3 and Figure A4).

The model’s performance was evaluated on individual cancers at test-time through simple inference and ensembling methods. We observed that the model’s predictive power improved when an ensemble of the checkpoints was employed (refer to Figure 3). However, for the four cancer types, TCGA-GBM, TCGA-LAML, TCGA-PRAD, and TCGA-TGCT, the model did not show significant predictive power. During the investigation, we observed that these datasets had non-admissible pairs in some of the data folds, i.e., all samples had censor value δ=0 (refer to Equation (A5)). In the case of TCGA-PRAD and TCGA-TGCT, the number of samples having δ=1 in the training/validation cohort was 12 and 3, respectively. To address the lack of predictive power, we fine-tuned the model for these datasets by using the stratified k-folds to offset the class-representation problem in the data folds. After searching for the optimal hyperparameters for fine-tuning, the model’s performance became significant (p<0.05) for three out of four datasets (refer to green box plots in Figure 3). Moreover, the TCGA-CDR consortium has already cautioned that OS is not recommended for TGCT owing to an insufficient number of events [9]. Figure 4 shows that tercile-based hazard scores sharply separate survival curves, spotlighting patients who need intensified care while identifying those with favorable prognoses. This clear gradient supports personalized treatment planning and evidence-guided follow-up to improve overall survival across diverse cohorts.

From a diagnostic perspective, cancer type classification is routinely studied for early detection and localization of tissue of origin [10]. The classification results in Table 2 illustrate the superior generalizability of the model’s predictive power to classify primary cancer types through the SeNMo encoder, despite it being primarily trained for predicting OS. Additionally, the detailed classification report across various cancer types reveals that the model consistently maintains high precision, recall, and F1-scores for almost all cancer types. Such metrics not only confirm the model’s effectiveness in accurately identifying the correct cancer class but also its reliability in replicating these results across different samples. This level of performance suggests the capability of the model to successfully learn high level representations from heterogenous, high-dimension, mutlivariate data stemming from complex molecular modalities such as gene expression, miRNA expression, somatic mutations, DNA methylation, and protein expression. Previous multi-omics classifiers often require modality-specific encoders or extensive feature engineering [11]; SeNMo achieves superior accuracy with a single backbone, simplifying deployment and maintenance.

Fine-tuning for TLS ratio prediction demonstrates that SeNMo’s latent space captures micro-environmental information not explicitly provided during pre-training. As shown in Figure 5, SeNMo’s ability to predict TLS ratios was evaluated on an unseen cohort of lung squamous cell carcinoma data from Moffitt Cancer Center. The comparison between manual TLS ratio annotations and SeNMo-predicted values showed no significant difference, indicating a high level of concordance between human annotations and model predictions. Violin plots depicting high vs. low TLS ratio groups—both for manual and SeNMo predictions—revealed significant separation, demonstrating the model’s robustness in distinguishing between biologically distinct TLS levels. Furthermore, KM survival curves for high vs. low TLS ratio groups revealed significant differences in survival outcomes, with stronger statistical significance observed for SeNMo-predicted data compared to manual annotations. The ability to reproduce pathologist TLS assessments and enhance survival stratification after minimal adaptation suggests that the model learns transferable features spanning tumor-intrinsic and immune contexts. This dual capacity is valuable for applications such as immunotherapy response prediction, where both tumor genetics and immune infiltration matter [12]. Overall, the results indicate that SeNMo can successfully generalize to new tasks and datasets, accurately predicting TLS ratios and offering valuable prognostic insights that could improve clinical decision-making.

It is imperative to mention here that MLP-based networks are very sensitive to catastrophic forgetting when presented with out-of-distribution data or when subjected to a different task [13]. We fine-tuned the SeNMo encoder for one public dataset (CPTAC-LSCC) and one internal dataset (Moffitt’s LSCC) [14,15]. In our simulations to fine-tune the model, we encountered the catastrophic forgetting phenomenon in SeNMo, where the model would fail to converge on both new datasets. This was more pronounced when a certain number of hidden layers were frozen, and the rest were trained with lower learning rates. We resorted to the option of unfreezing all the layers of the encoder and fine-tuning the model with a very small learning rate ($4\times 10^{-5}$), high weight decay and dropout (0.35), and just 10 epochs. This method worked, and the model showed significant performance on the out-of-distribution datasets.

SeNMo’s unified architecture enables simultaneous estimation of survival risk, tumor origin, and immune context from the same patient profile, streamlining molecular decision-support workflows. In settings with incomplete data, which is common in routine oncology practice, the model degrades gracefully, allowing actionable output even when only transcriptomics are available. The strong external performance on CPTAC-LSCC and Moffitt-LSCC shows that the learned representation transfers across sequencing platforms and institutional pipelines after brief calibration. Because of the modular structure of SeNMo, its data-ingestion layer can accept any external normalization, e.g., RNA-seq matrices processed with limma-voom [16] or the DESeq2 variance-stabilizing transform [17] can be routed into the same encoder, while the other omics tracks retain the default log+z pipeline. This plug-and-play design lets researchers incorporate task-specific preprocessing or differential-expression outputs without modifying the network architecture, enabling seamless extension of the framework to future modality-specific workflows.

There are some limitations to this work that warrant discussion. First, despite robust overall performance, four cancer types exhibited low baseline C-indices and one (TGCT) remained refractory to fine-tuning, indicating that tumor-specific molecular peculiarities can still elude the shared model. Second, the external validation cohorts were confined to one internal and one public data cohort; broader evaluation across additional histologies will further substantiate generalizability. Third, TLS labels were derived from image-analysis pipelines, incorporating an image-based deep learning encoder would potentially automate this process and reduce the measurement noise. The research community is invited to advance this work in future; some avenues of exploration include (i) contrastive pre-training with synthetic modality dropout to enhance robustness, (ii) incorporation of spatial transcriptomics and digital pathology features to better characterize micro-environmental heterogeneity, (iii) inclusion of other molecular data modalities such as mRNA, CNVs/CNAs by adapting the SeNMo’s encoder to variable input size and adding a fine-tuning head for downstream prediction, and (iv) prospective validation within molecular tumor boards to quantify clinical impact on treatment decisions.

## 4. Materials and Methods

Figure 1 presents the outline of this work that includes the data types and characteristics (Figure 1A,B), preprocessing and model development (Figure 1C,D), and the study design and simulations (Figure 1E). Below, we discuss each of these steps in detail.

### 4.1. Data Acquisition

TCGA houses one of the largest collections of high-dimensional multi-omics datasets, comprising over 20,500 individual tumor samples from 33 different cancer types [18]. The available data includes high-throughput RNA sequencing (RNA-Seq), DNA sequencing (DNA-Seq), microRNA sequencing (miRNA-Seq), single nucleotide variants, copy number variations, DNA methylation, and reverse-phase protein array (RPPA) data [18]. Building cohorts from this diverse data, spanning multiple formats, modalities, and systems, presents significant challenges. To curate and establish patient cohorts, we utilized our previously developed Multimodal Integration of Oncology Data System (MINDS), a metadata framework designed to fuse data from publicly available sources like TCGA-GDC and UCSC Xena Portal into a machine learning-ready format [3,18,19]. MINDS is freely accessible to the cancer research community and has been integrated into the SeNMo framework to enhance its usability and benefit to researchers. For training, validation, and testing, we used pan-cancer data from TCGA and Xena, covering 33 cancer types. We further fine-tuned the model using data from the CPTAC-LSCC [14] and Moffitt’s LSCC datasets [15] to evaluate the generalizability and transfer learning capabilities of SeNMo. The details of clinical data characteristics are given in Table 3.

### 4.2. Data Modalities

From the 13 available modalities present in each cancer dataset, we selected gene expression (RNAseq), DNA methylation, miRNA stem-loop expression, RPPA data, DNA mutation, and clinical data. These modalities were chosen based on their frequent use in cancer studies due to their direct relevance to the fundamental processes of cancer progression, as well as their diagnostic and prognostic capabilities [20,21]. Furthermore, these selected modalities provide robust predictive and prognostic information, and their integration gives a holistic view of a tumor’s multi-omic profile [20,21,22]. Importantly, each modality had a consistent number of features across all cancer types, which facilitated the development of a standardized data preprocessing pipeline for pan-cancer studies. TCGA-GDC multi-omics data includes DNA methylation (485,576 CpG sites, beta values 0–1), gene expression (RNAseq) (60,483 features, HTseq-FPKM values log-transformed after incrementing by one, with high expression > 1000 and low between 0.5–10), miRNA stem-loop expression (1880 features, log-transformed values), protein expression (RPPA data profiling 500 proteins, normalized via log transformation and median centering), DNA mutation (18,090 mutational features from MAF files summarizing somatic mutations), and clinical data (patient covariates like age, gender, race, and cancer stage, essential for prognosis and treatment response). For details on each modality, see Appendix A.

### 4.3. Pre-Processing

Multiomics data integrates diverse biological data modalities such as genomics, transcriptomics, proteomics, and metabolomics to study complex diseases like cancer but requires extensive preprocessing due to the *big P, small n* problem—high dimensionality (P) with limited samples (n) [23]. The pan-cancer multi-omics data comes with intra- and inter-dataset correlations, heterogeneous measurement scales, missing values, technical variability, and other background noise. Preprocessing challenges include data heterogeneity, large volume and complexity, quality and variability, missing values, and lack of standardization across studies. Effective preprocessing involves normalization, scaling, handling missing data, dimensionality reduction, data annotation, metadata inclusion, and selecting appropriate integration techniques to ensure data is machine learning-ready. Addressing these challenges requires interdisciplinary expertise, including bioinformatics, statistics, and domain-specific knowledge. Here, we describe the preprocessing steps used across molecular data modalities.

Remove NaNs: First, we removed the features that had NaNs across all the samples. This reduced the dimension, removed noise, and ensured continuous-numbered features to work with.Drop constant features: Next, constant/quasi-constant features with a threshold of 0.998 were filtered out using Feature-engine, a Python library for feature engineering and selection [24]. The threshold was selected so as to remove only the features with no expression at all across every sample and also features that had high noise contents, since the expression value was the same across every sample.Remove duplicates features: Next, duplicate features between genes were identified that contained the same values across two seperate genes, and one of the genes was kept. This may reveal gene–gene relationships between the two genes stemming from an up-regulation pathway or could simply reflect noise.Remove colinear features: Next, we filtered the features having low variance (≈0.25) because the features having high variance hold the maximum amount of information [25]. We used the VarianceThreshold feature selector of scikit learn library that removes low-variance features based on the defined threshold [26]. We chose a threshold for each data modality so that the resulting features have matching dimensions, as shown in Figure 1D.Remove low-expression genes: The gene expression data originally contained 60,483 features, with FPKM transformed numbers ranging from 0 to 12. Roughly 30,000 features remained after the above-mentioned preprocessing steps, which was still a very high number of features. High expression values reveal important biological insights due to an indication that a certain gene product is transcribed in large quantities, revealing their relevance compared to low expression values [27]. Although there is no well-defined consensus on the selection of cut-off value, existing practices involve keeping high-variance and high-expression values [28,29,30,31]. Based on evidence from existing literature and our empirical analysis on the resultant feature dimension [30,31], features containing an expression value greater than 7 (127 FPKM value) were kept for our simulations.Handle missing features: We handled missing features at two levels of data integration. First, for the features within each modality and cancer type, the missing values were imputed with the mean of the samples for that feature. This resulted in the full-length feature vector for each sample. Second, across different cancers and modalities, we padded the missing features with zeros. In deep learning, the zero imputation technique shows the best performance compared to other imputation techniques and deficient data removal techniques [32,33,34].

### 4.4. Features Integration

After preprocessing, the data is integrated across cancers and modalities, generating two views: one by unifying features across all patients within each modality, and second by combining all modalities. Feature dimension post-preprocessing includes DNA methylation (reduced from 485,576 to ∼4500 per patient, unionized to 52,396), gene expression (60,483 to ∼3000 per patient, unionized to 8794), miRNA expression (1880 to ∼1400 per patient, unionized to 1730), protein expression (487 to 472), and DNA mutation (18,090 to 17,253). Details of these feature reductions and integrations are summarized in Table 4. In addition to integrating these multi-omics data, we also integrate four clinical data features, including age, gender, race, and cancer stage. These features are added to the combined multi-omics feature through concatenation, resulting in the per-sample feature dimension of 80,697. Mathematically, the integration is explained below.(1)$$V_{m}=\left\{\begin{array}{cc} \overset{n}{\underset{i=1}{⋃}}{\tilde{v}}_{i} & \mathrm{if}\;{\tilde{v}}_{i}\mathrm{varies}\;\mathrm{by}\;\mathrm{cancer}\;\mathrm{type}\;\mathrm{or}\;\mathrm{modality},\\ \tilde{v} & \mathrm{otherwise}. \end{array}\right.$$

Finally, the union of all $V_{m}$ across different modalities results in the total pan-cancer, multimodal feature vector $V_{c}\in R^{80,697}$, expressed as:(2)$$V_{c}=\overset{M}{\underset{m=1}{⋃}}V_{m}$$

### 4.5. Clinical Endpoints

To assess the performance of the SeNMo framework, we selected diagnostic and prognostic endpoints. The first end-point is OS, which is treated as a regression task. The second is the prediction of primary cancer type, formulated as a 33-class classification task. The third endpoint is TLS ratio prediction, also a regression task.

#### 4.5.1. Primary Cancer Type

Diagnosing primary cancer type based on biological and clinical features is critical for accurate treatment selection, improving patient outcomes, and enabling personalized therapies [35,36]. Since treatments and prognoses vary across cancer types, precise classification supports tailored interventions, follow-up care, and early recurrence detection. High classification accuracy enhances clinical decision-making, ultimately benefiting patient survival and quality of life. The distribution of patient samples across 33 cancer types is shown in Figure 1A.

#### 4.5.2. Overall Survival (OS)

Cancer prognosis through survival outcomes is a standard approach for biomarker discovery, patient stratification, and assessing therapeutic response [5]. Advances in statistical survival models and deep learning have improved OS prediction by integrating molecular and clinical data [37,38]. In this study, we analyze clinical, demographic, and genomic factors to assess their correlation with patient survival, implementing OS prediction as a regression task in days. Survival data includes time-to-event records, accounting for right censoring when exact survival times are unknown. Figure 1B illustrates survival times since cancer diagnosis for the pan-cancer dataset. Each patient’s outcome is characterized by two variables: a censoring indicator, also known as the vital status, and the observed time $T=\min(T_{s},T_{\delta })$, where $T_{s}$ represents the true survival time and $T_{\delta }$ is the censoring time, $\left\{T_{s}\leq T_{\delta }\right\}$ [7]. The survival function, which describes the probability that a patient will survive beyond a specified time *t*, is given by:(3)F(t)=P{T>t}.

Additionally, the hazard function provides insight into the risk of an event occurring at a particular time, given survival up to that point. It represents the instantaneous rate of events (e.g., death) occurring at a specific time, conditional on having survived to that time. The hazard function h(t) is mathematically defined as the ratio of the probability of the event occurring in a short interval around *t* to the probability of surviving beyond *t*:(4)$$h\left(t\right)=\lim_{\Delta t\to 0}\frac{P(t\leq T<t+\Delta t\;|\;T\geq t)}{\Delta t},$$
where h(t) is the hazard function at time *t*, *T* is the survival time, P(t≤T<t+Δt|T≥t) is the conditional probability that the event occurs in the time interval [t,t+Δt) given that survival time is greater than or equal to *t*, and Δt represents an infinitesimally small time interval. Based on survival data, the hazard function describes the instantaneous risk of experiencing the event of interest at any given time. In our study, right-censoring was defined as censor δ=1 in case of an event (e.g., death), and 0 otherwise.

#### 4.5.3. Tertiary Lymphoid Structures (TLS) Ratio

TLSs are immune cell aggregates resembling secondary lymphoid organs, forming in inflamed tissues, including cancers, and are associated with improved survival and immunotherapy response [39,40]. The TLS ratio (segmented TLS area/total tissue area) serves as a prognostic biomarker, influencing clinical decision-making. Automated TLS segmentation models have shown high accuracy across multiple cancers [39,40]. In this study, whole slide images of H&E and CD20-stained sections were analyzed using Visiopharm 2022.03. Visiopharm Tissuealign co-registered serial images, tumor and non-tumor regions were manually segmented, and TLSs were detected via thresholding, followed by manual review and feature extraction by an experienced image analysis technician under pathologist supervision. The tissue-level TLS ratio was calculated as the ratio of the segmented TLS area to the total tissue area. The initial sample count for data was N=108. As shown in Figure 5a, out of the 108 patients, 7 had missing TLS labels and were excluded from the study.

### 4.6. SeNMo Deep Learning Model

In scenarios involving hundreds or thousands of features with relatively few training samples, feedforward networks often face the risk of overfitting [5]. Unlike CNNs, weights in feedforward networks are shared, making them vulnerable to training instabilities caused by perturbations and regularization techniques such as stochastic gradient descent and dropout. CNNs, on the other hand, struggle to handle high-dimensional, low-sample data due to the spatial invariance assumption, fixed input size, and inefficiencies in managing multi-omics data sparsity. Transformer-based models are also suboptimal for high-dimensional, low-sample data, as they rely heavily on attention mechanisms tailored for predicting sequential patterns, which fails when dealing with highly sparse molecular data. To address the challenges of overfitting and instability in high-dimensional, low-sample-size multi-omics data, we drew inspiration from self-normalizing networks introduced by Klambauer et al. [41]. Self-normalizing neural networks are particularly suited for high-dimensional datasets with limited samples, a characteristic that makes them highly relevant for multi-omics analysis. The SeNMo architecture is based on stacked layers of self-normalizing neural networks, as detailed below.

SeNMo comprises stacked blocks of self-normalizing neural network layers, where each block includes a linear unit, a Scaled Exponential Linear Unit (SELU) activation, and Alpha-Dropout. These components enable high-level abstract representations while keeping neuron activations close to zero mean and unit variance [41]. The linear unit is equivalent to a “fully connected” or MLP layer commonly used in traditional neural network architectures. Klambauer et al. demonstrated through the Banach fixed-point theorem that activations with close proximity to zero mean and unit variance, propagating through numerous network layers, will ultimately converge to zero mean and unit variance [41]. SELU activations, an alternative to traditional rectified linear unit activations, offer a self-normalizing effect, ensuring activations converge to zero mean and unit variance regardless of the input distribution. The SELU activation function is expressed mathematically as:(5)$$\mathrm{SELU}\left(x\right)=\lambda \left\{\begin{array}{cc} x & \mathrm{if}\;x>0\\ \alpha (e^{x}-1) & \mathrm{if}\;x\leq 0, \end{array}\right.$$
where λ is a scaling factor (typically set to 1.05071) and α is the negative scale factor (typically set to 1.6733).

Dropout, a regularization method that randomly sets a fraction of input units to zero during training, prevents overfitting. Alpha-Dropout, a modified version of traditional dropout, is designed to maintain the self-normalizing property of SELU activations. It applies a dropout mask during training, scaled to ensure the mean and variance of activations remain stable. The scaling factor is computed based on the dropout rate and the SELU parameters (λ and α). Alpha-Dropout is mathematically defined as:(6)$$\mathrm{Alpha}-\mathrm{dropout}\left(x\right)=\frac{x-\mu (x)}{std(x)}\times mask+\mu \left(x\right),$$
where *x* is the input activation, μ(x), std(x) are the mean and standard deviation of the input activation, respectively, and mask is a binary mask generated with the specified dropout rate.

Together, SELU activations and Alpha-Dropout ensure that SeNMo blocks maintain stable mean and variance across network layers, facilitating more reliable training and better generalization performance. Additionally, these mechanisms help mitigate training instabilities related to vanishing or exploding gradients in feedforward networks. Our network architecture consists of seven fully connected hidden layers, each followed by SELU activation and Alpha-Dropout. The final fully connected layer is used to learn a latent representation of each sample, termed as the patient embedding $x\in R^{48}$.

### 4.7. Training and Evaluation

#### 4.7.1. Data Splits

For the OS task, the pan-cancer data was randomly divided into the training-validation set (80%) and the hold-out test set (20%) for each cancer type. The pan-cancer training was carried out by combining the training-validation cohort of all 33 cancer types and adopting the 10-fold cross-validation with the 80–20% division of samples. The training-validation cohort has 11,050 patients, each having $R^{80,697}$ features, comprising the six modalities mentioned earlier. For the evaluation/testing of the trained model, the inference data was created by combining the held-out test set from all 33 cancer types, resulting in 2754 patients. We tested the optimal hyperparameters of our trained model to train different combinations of the pan-cancer data modalities. We call these 1-modal, 3-modal (gene expression, DNA methylation, miRNA expression), 4-modal (3+ protein expression), 5-modal (4+ DNA mutation), and 6-modal (all modalities) cohorts. Although our initial model was trained on all 6 modalities, these experiments aim to see how the model performs on each of these pan-cancer cohorts where one or more of the data modalities is missing.

#### 4.7.2. Evaluation

We evaluate SeNMo’s performance with the quantitative and statistical metrics common for survival outcome prediction and classification. For survival analysis, we evaluated the model using the C-index. For the primary cancer type classification, we generate the classification report comprising average accuracy, average precision, recall, F1-score, confusion matrix, and scatter plot. For the TLS Ratio, we employed Huber Loss. We utilized the log-rank test to determine if the survival predictions were statistically significantly different. For details on each evaluation metric, see Appendix B.

### 4.8. Study Design

An overview of the various simulations conducted to evaluate the capabilities of the SeNMo model across different learning regimes, tasks, and datasets is shown in Figure 1E. The simulations included multiple learning regimes, each designed to assess the model’s adaptability, generalizability, and robustness on diagnostic and prognostic tasks. The baseline model was initially trained on the TCGA dataset comprising 33 different cancer types for OS prediction. The subsequent learning regimes explored different data variations and tasks, which we call out-of-distribution simulations because the model had not encountered such data/task in baseline learning. These scenarios included OS prediction on both seen and unseen datasets, as well as tasks such as primary cancer type classification on seen data and TLS ratio prediction on unseen data.

## 5. Conclusions

In this study, we introduced SeNMo, a deep learning model specifically designed for multi-omics data analysis. SeNMo offers an end-to-end framework that unifies prognostic, diagnostic, and immune micro-environment modeling within a single, modality-agnostic network. The results highlight that self-normalizing architectures alleviate the optimization challenges posed by high-dimensional, sparse multi-omics data, enabling reliable performance even in incomplete or shifted domains. By making SeNMo and its derived patient embeddings publicly available, we aim to facilitate further research and innovation in personalized cancer care, underscoring the transformative potential of multi-omics approaches in the fight against cancer.

## Figures and Tables

**Figure 1 ijms-26-07358-f001:**
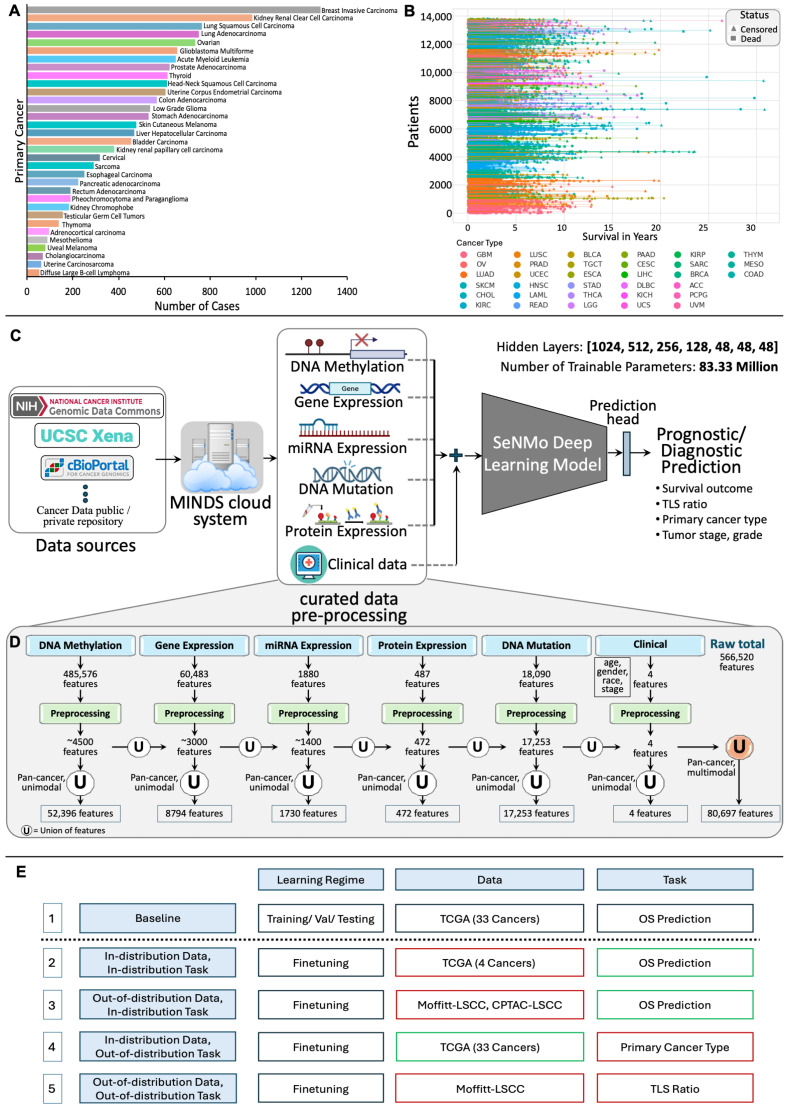
(**A**): Number of cases for each cancer type, highlighting variation in sample sizes across cancers. (**B**): Patient survival in years, with each line representing an individual patient, marked as censored or dead, and grouped by cancer type. (**C**): Overview of the SeNMo framework. The data from public and private sources are collected using Multimodal Integration of Oncology Data System (MINDS) [3] and curated to develop the multiomics dataset. The learned model weights are later used for different downstream prognostic and diagnostic tasks. (**D**): Features processing pipeline for six data types. Initial modality features reduced through preprocessing are unified at pan-cancer level to yield unimodal pan-cancer feature sets, which are further unified across modalities, resulting in a multiomics matrix of 80,697 features. (**E**): Study design and simulations for SeNMo across different learning regimes, datasets, and tasks. The baseline model (row 1) is first trained on TCGA data with 33 cancer types for overall survival prediction. Rows 2–5 represent in-distribution data/task (same as baseline) and out-of-distribution data/task (different from baseline) variations: red-bordered boxes indicate a change from the baseline, while green-bordered boxes align with the baseline.

**Figure 2 ijms-26-07358-f002:**
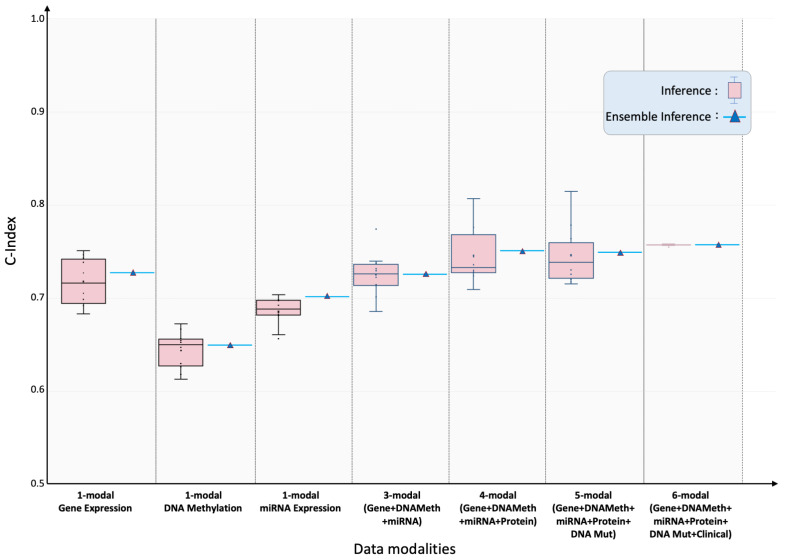
Pan-cancer concordance indices for overall-survival prediction under incremental modality integration. SeNMo model was trained and evaluated, using different combinations of data modalities, on 80% of the total data, and evaluated on the 20% held-out test set. Box-plots summarize ten-fold cross-validation results for models ingesting 1-, 3-, 4-, 5- and 6-modal input configurations. As the number of modalities increased, the model’s performance improved, confirming additive prognostic value from each modality. All scores surpass the 95% significance threshold (log-rank p<0.05).

**Figure 3 ijms-26-07358-f003:**
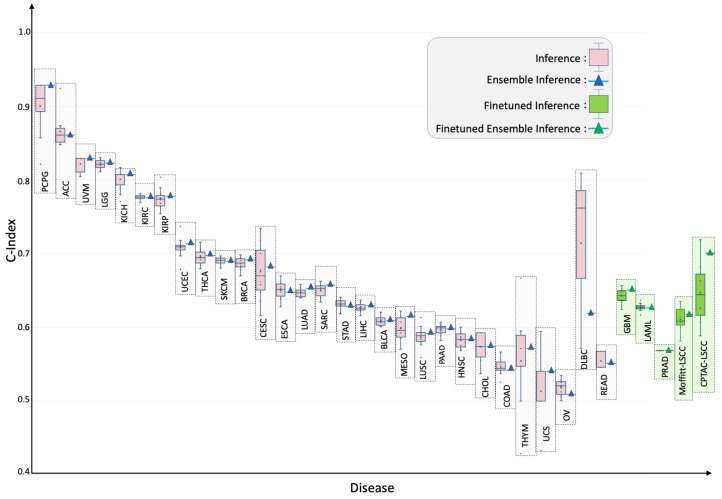
Tumor-specific concordance indices for SeNMo across 32 cancer types and two external lung-squamous cohorts. Pink box-plots depict zero-shot predictions on each held-out TCGA subset; green box-plots show outcomes after a brief ten-epoch fine-tune where baseline performance was not significant or the cohort was unseen (CPTAC-LSCC, Moffitt-LSCC). Ensemble estimates are indicated by triangles. Although trained on pan-cancer cohort, SeNMo effectively captures survival times across individual cancers. Fine-tuning improved performance for cases with initially low predictive significance, and markedly improved external-cohort accuracy, illustrating SeNMo’s capacity for rapid domain adaptation.

**Figure 4 ijms-26-07358-f004:**
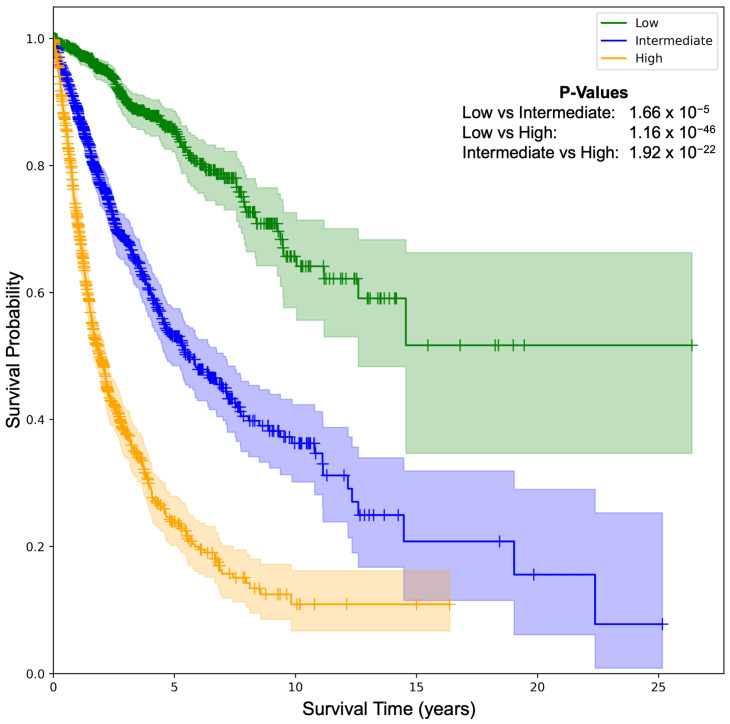
Kaplan–Meier (KM) survival curves illustrating SeNMo-driven risk stratification on the pan-cancer data. Patients are partitioned into low, intermediate, and high-risk terciles using the 33rd and 66th percentiles of predicted hazard scores. Survival trajectories diverge markedly across strata. The *p*-values from logrank test for Low vs. Intermediate: $1.66\times 10^{-5}$, Low vs. High: $1.16\times 10^{-46}$, and Intermediate vs. High: $1.92\times 10^{-22}$. The shaded areas around each curve depict the 95% confidence intervals.

**Figure 5 ijms-26-07358-f005:**
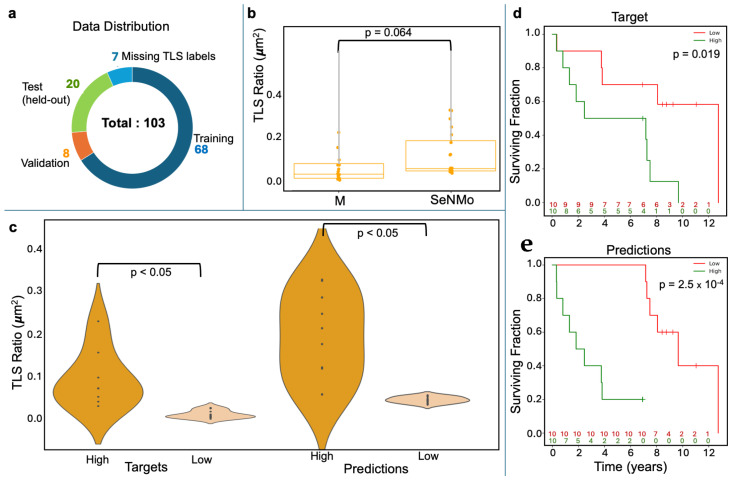
Tertiary Lymph Structures (TLSs) ratio predictions by SeNMo. (**a**) Dataset distribution, dividing 103 samples into training, validation, test, and excluding those missing TLS data. (**b**) Box plots compare TLS ratios between manual annotations (M) and SeNMo predictions, showing no significant difference (p=0.064). (**c**) Violin plot of TLS Ratio annotations for high vs. low groups thresholded by the median value, with a significant difference (p<0.05). (**d**) Violin plot of SeNMo’s TLS Ratio predictions, also showing significant separation between high and low groups (p<0.05). (**e**) Kaplan–Meier survival curves with significant survival differences, comparing high vs. low TLS ratios for both annotations (top, p=0.019) and SeNMo predictions (bottom, $p=2.5\times 10^{-4}$).

**Table 1 ijms-26-07358-t001:** C-Index for test and ensemble inference across cancer types.

Cancer Type	C-Index {Test, Ensemble}	Cancer Type	C-Index {Test, Ensemble}
TCGA-PCPG	{0.900, 0.929}	TCGA-DLBC	{0.714, 0.619}
TCGA-ACC	{0.866, 0.861}	TCGA-MESO	{0.599, 0.615}
TCGA-UVM	{0.822, 0.829}	TCGA-LUSC	{0.588, 0.592}
TCGA-LGG	{0.821, 0.823}	TCGA-PAAD	{0.597, 0.598}
TCGA-KICH	{0.801, 0.807}	TCGA-HNSC	{0.583, 0.583}
TCGA-KIRC	{0.777, 0.776}	TCGA-CHOL	{0.574, 0.574}
TCGA-KIRP	{0.775, 0.778}	TCGA-COAD	{0.546, 0.542}
TCGA-UCEC	{0.708, 0.713}	TCGA-THYM	{0.555, 0.571}
TCGA-THCA	{0.696, 0.698}	TCGA-UCS	{0.514, 0.541}
TCGA-SKCM	{0.691, 0.689}	TCGA-OV	{0.518, 0.509}
TCGA-BRCA	{0.687, 0.692}	TCGA-READ	{0.550, 0.551}
TCGA-CESC	{0.676, 0.682}	TCGA-GBM	{0.642, 0.650} *
TCGA-ESCA	{0.650, 0.648}	TCGA-LAML	{0.627, 0.626} *
TCGA-LUAD	{0.647, 0.653}	TCGA-PRAD	{0.541, 0.542} *
TCGA-SARC	{0.650, 0.658}	TCGA-STAD	{0.631, 0.628}
TCGA-LIHC	{0.627, 0.629}	TCGA-BLCA	{0.609, 0.609}
		TCGA-TGCT	{0.123, 0.091 } **

* after few-shots fine-tuning because inference on baseline was not statistically significant (p>0.05); ** failure case.

**Table 2 ijms-26-07358-t002:** Classification report of different diseases with precision, recall, F1-score, and support metrics across 33 cancer types.

Primary Site	Data	Class	Precision	Recall	F1-Score	Support
Adrenocortical	TCGA-ACC	0	1	1	1	16
Bladder	TCGA-BLCA	1	1	1	1	89
Breast	TCGA-BRCA	2	1	1	1	245
Cervical	TCGA-CESC	3	1	1	1	62
Bile Duct	TCGA-CHOL	4	1	1	1	8
Colon	TCGA-COAD	5	0.99	1	1	108
Large B-cell Lymphoma	TCGA-DLBC	6	1	1	1	10
Esophageal	TCGA-ESCA	7	1	1	1	44
Glioblastoma	TCGA-GBM	8	0.94	1	0.97	48
Head and Neck	TCGA-HNSC	9	1	1	1	121
Kidney Chromophobe	TCGA-KICH	10	1	1	1	12
Kidney Clear Cell Carcinoma	TCGA-KIRC	11	1	1	1	146
Kidney Papillary Cell Carcinoma	TCGA-KIRP	12	1	1	1	71
Acute Myeloid Leukemia	TCGA-LAML	13	1	1	1	22
Lower Grade Glioma	TCGA-LGG	14	1	1	1	107
Liver	TCGA-LIHC	15	1	1	1	87
Lung Adenocarcinoma	TCGA-LUAD	16	1	1	1	126
Lung Squamous Cell Carcinoma	TCGA-LUSC	17	0.98	0.99	0.99	113
Mesothelioma	TCGA-MESO	18	1	1	1	18
Ovarian	TCGA-OV	19	1	1	1	104
Pancreatic	TCGA-PAAD	20	1	1	1	121
Pheochromocytoma & Paraganglioma	TCGA-PCPG	21	1	1	1	38
Prostate	TCGA-PRAD	22	1	1	1	117
Rectal	TCGA-READ	23	1	1	1	23
Sarcoma	TCGA-SARC	24	1	1	1	52
Skin Cutaneous Melanoma	TCGA-SKCM	25	1	1	1	96
Stomach	TCGA-STAD	26	1	1	1	121
Testicular	TCGA-TGCT	27	1	1	1	25
Thyroid	TCGA-THCA	28	1	1	1	97
Thymoma	TCGA-THYM	29	1	1	1	24
Endometrioid	TCGA-UCEC	30	1	1	1	193
Uterine Carcinosarcoma	TCGA-UCS	31	1	1	1	11
Uveal melanomas	TCGA-UVM	32	1	1	1	16
Weighted Avg			**1**	**1**	**1**	**2360**

**Table 3 ijms-26-07358-t003:** Summary of patient characteristics for pan-cancer data in this study.

Cancer Type	Age (Mean ± SD)	Gender (M/F)	Race (White/Asian/Black/NA/ American Indian/Alaska)	Stage (0/I/IA/IB/IC/II/IIA/IIB/IIC/III/IIIA/IIIB/IIIC/IV/IVA/IVB/IVC/NA)
TCGA-ACC	47.46 ± 16.20	33/62	79/3/1/12/0	0/9/0/0/0/46/0/0/0/20/0/0/0/17/0/0/0/3
TCGA-BLCA	67.92 ± 10.39	326/121	363/43/23/18/0	0/3/0/0/0/136/0/0/0/159/0/0/0/148/0/0/0/1
TCGA-BRCA	57.94 ± 13.11	13/1247	915/59/198/87/1	0/114/94/7/0/6/404/307/0/2/176/30/74/22/0/0/0/24
TCGA-CESC	48.04 ± 13.70	0/304	211/19/32/30/9	0/0/0/0/0/0/0/0/0/0/0/0/0/0/0/0/0/304
TCGA-CHOL	64.37 ± 12.21	30/32	55/3/3/1/0	0/30/0/0/0/16/0/0/0/5/0/0/0/2/3/6/0/0
TCGA-COAD	66.93 ± 12.67	288/251	261/11/67/198/2	0/87/1/0/0/46/150/13/2/26/9/69/47/56/18/3/0/12
TCGA-DLBC	56.76 ± 13.68	24/27	32/18/1/0/0	0/0/0/0/0/0/0/0/0/0/0/0/0/0/0/0/0/51
TCGA-ESCA	64.22 ± 12.11	208/41	162/46/6/35/0	0/14/9/7/0/1/56/43/0/41/16/10/9/7/6/0/0/30
TCGA-GBM	57.74 ± 14.32	399/250	547/13/53/36/0	0/0/0/0/0/0/0/0/0/0/0/0/0/0/0/0/0/649
TCGA-HNSC	61.02 ± 11.92	443/168	522/12/58/17/2	0/29/0/0/0/93/0/0/0/97/0/0/0/0/302/13/1/76
TCGA-KICH	51.61 ± 14.12	99/83	154/6/19/3/0	0/75/0/0/0/59/0/0/0/34/0/0/0/14/0/0/0/0
TCGA-KIRC	60.67 ± 11.95	641/338	876/16/73/14/0	0/475/0/0/0/102/0/0/0/237/0/0/0/161/0/0/0/4
TCGA-KIRP	61.98 ± 12.20	278/98	275/6/75/16/4	0/219/0/0/0/25/0/0/0/77/0/0/0/21/0/0/0/34
TCGA-LAML	54.82 ± 15.87	345/281	564/8/49/5/0	0/0/0/0/0/0/0/0/0/0/0/0/0/0/0/0/626
TCGA-LGG	42.71 ± 13.32	293/240	492/8/22/10/1	0/0/0/0/0/0/0/0/0/0/0/0/0/0/0/0/533
TCGA-LIHC	60.44 ± 13.71	305/158	255/168/25/14/1	0/211/0/0/0/105/0/0/0/6/78/12/11/2/1/3/0/34
TCGA-LUAD	65.20 ± 10.08	329/399	580/14/84/48/2	0/7/194/195/0/2/67/103/0/0/101/12/0/37/0/0/0/10
TCGA-LUSC	67.28 ± 8.62	548/204	530/12/47/163/0	0/4/127/243/0/4/87/138/0/3/94/33/0/12/0/0/0/7
TCGA-MESO	63.01 ± 9.78	70/16	84/1/1/0/0	0/7/2/1/0/15/0/0/0/45/0/0/0/16/0/0/0/0
TCGA-OV	59.60 ± 11.44	0/731	626/25/43/33/3	0/0/0/0/0/0/0/0/0/0/0/0/0/0/0/0/731
TCGA-PAAD	64.87 ± 11.36	123/99	195/13/8/6/0	0/1/6/15/0/0/36/148/0/6/0/0/0/7/0/0/0/3
TCGA-PCPG	47.02 ± 15.15	84/105	157/7/20/4/1	0/0/0/0/0/0/0/0/0/0/0/0/0/0/0/0/189
TCGA-PRAD	60.93 ± 6.80	623/0	510/13/81/18/1	0/0/0/0/0/0/0/0/0/0/0/0/0/0/0/0/623
TCGA-READ	63.83 ± 11.85	98/80	90/1/7/80/0	0/37/0/0/0/7/40/2/1/6/7/25/14/21/7/0/0/11
TCGA-SARC	60.70 ± 14.38	129/158	253/5/20/9/0	0/0/0/0/0/0/0/0/0/0/0/0/0/0/0/0/287
TCGA-SKCM	57.84 ± 15.41	289/174	441/12/1/9/0	6/30/18/30/0/39/18/28/61/44/16/46/68/23/0/0/0/36
TCGA-STAD	65.44 ± 10.53	320/179	311/108/15/64/0	0/1/21/46/0/37/54/71/0/4/88/67/39/47/0/0/0/24
TCGA-TGCT	31.87 ± 9.19	139/0	124/4/6/5/0	0/69/26/11/0/4/6/1/1/2/1/6/5/0/0/0/0/7
TCGA-THCA	47.17 ± 15.83	166/448	413/59/35/106/1	0/350/0/0/0/64/0/0/0/134/0/0/0/4/52/0/8/2
TCGA-THYM	58.12 ± 13.00	72/66	115/13/8/2/0	0/0/0/0/0/0/0/0/0/0/0/0/0/0/0/0/138
TCGA-UCEC	63.74 ± 11.06	0/588	402/21/120/32/4	0/0/0/0/0/0/0/0/0/0/0/0/0/0/0/0/588
TCGA-UCS	70.07 ± 9.24	0/61	50/1/9/1/0	0/0/0/0/0/0/0/0/0/0/0/0/0/0/0/0/61
TCGA-UVM	61.65 ± 13.95	45/35	55/0/0/25/0	0/0/0/0/0/0/12/27/0/0/25/10/1/4/0/0/0/1
Moffitt-LSCC	69.14 ± 8.34	72/36	105/0/3/0/0	0/0/24/25/0/0/31/15/0/0/12/1/0/0/0/0/0/0

**Table 4 ijms-26-07358-t004:** Feature reduction summary of pan-cancer data.

Data (TCGA-GDC)	Primary Site	Cases	miRNA Exprn		DNA Methyl		Gene Exprn		Protein Exprn		DNA Mut	
			Before	After	Before	After	Before	After	Before	After	Before	After
TCGA-DLBC	Large B-cell Lymphoma	51	1880	1060	485,576	4396	60,483	850	487	472	18,090	17,253
TCGA-UCS	Uterine Carcinosarcoma	61	1880	1101	485,576	4632	60,483	1231	487	472	18,090	17,253
TCGA-CHOL	Bile Duct	62	1880	967	485,576	4479	60,483	1261	487	472	18,090	17,253
TCGA-UVM	Uveal melanomas	80	1880	1162	485,576	4019	60,483	772	487	472	18,090	17,253
TCGA-MESO	Mesothelioma	86	1880	1158	485,576	4372	60,483	1278	487	472	18,090	17,253
TCGA-ACC	Adrenocortical	95	1880	1110	485,576	4454	60,483	1304	487	472	18,090	17,253
TCGA-THYM	Thymoma	138	1880	1245	485,576	4609	60,483	1337	487	472	18,090	17,253
TCGA-TGCT	Testicular	139	1880	1290	485,576	4762	60,483	1343	487	472	18,090	17,253
TCGA-READ	Rectal	178	1880	1314	485,576	4077	60,483	1547	487	472	18,090	17,253
TCGA-KICH	Kidney Chromophobe	182	1880	1089	485,576	4333	60,483	1107	487	472	18,090	17,253
TCGA-PCPG	Pheochromocytoma and Paraganglioma	189	1880	1251	485,576	4550	60,483	1216	487	472	18,090	17,253
TCGA-PAAD	Pancreatic	222	1880	1308	485,576	4518	60,483	1567	487	472	18,090	17,253
TCGA-ESCA	Esophageal	249	1880	1300	485,576	4192	60,483	1684	487	472	18,090	17,253
TCGA-SARC	Sarcoma	287	1880	1235	485,576	4467	60,483	2490	487	472	18,090	17,253
TCGA-CESC	Cervical	304	1880	1405	485,576	4167	60,483	2017	487	472	18,090	17,253
TCGA-KIRP	Kidney Papillary Cell Carcinoma	376	1880	1297	485,576	4078	60,483	1798	487	472	18,090	17,253
TCGA-SKCM	Skin Cutaneous Melanoma	436	1880	1426	485,576	4427	60,483	2488	487	472	18,090	17,253
TCGA-BLCA	Bladder	447	1880	1361	485,576	4483	60,483	2751	487	472	18,090	17,253
TCGA-LIHC	Liver	463	1880	1336	485,576	4023	60,483	2017	487	472	18,090	17,253
TCGA-STAD	Stomach	499	1880	1397	485,576	4196	60,483	2354	487	472	18,090	17,253
TCGA-LGG	Lower Grade Glioma	533	1880	1287	485,576	4193	60,483	1560	487	472	18,090	17,253
TCGA-COAD	Colon	539	1880	1460	485,576	4671	60,483	1931	487	472	18,090	17,253
TCGA-UCEC	Endometrioid	588	1880	1414	485,576	4424	60,483	2849	487	472	18,090	17,253
TCGA-HNSC	Head and Neck	611	1880	1428	485,576	4358	60,483	2059	487	472	18,090	17,253
TCGA-THCA	Thyroid	614	1880	1369	485,576	4160	60,483	1432	487	472	18,090	17,253
TCGA-PRAD	Prostate	623	1880	1334	485,576	4006	60,483	1635	487	472	18,090	17,253
TCGA-LAML	Acute Myeloid Leukemia	626	1880	1140	485,576	4415	60,483	1032	487	472	18,090	17,253
TCGA-GBM	Glioblastoma	649	1880	1023	485,576	4076	60,483	1206	487	472	18,090	17,253
TCGA-LUAD	Lung Adenocarcinoma	728	1880	1360	485,576	4480	60,483	2562	487	472	18,090	17,253
TCGA-OV	Ovarian	731	1880	1430	485,576	4254	60,483	2116	487	472	18,090	17,253
TCGA-LUSC	Lung Squamous Cell Carcinoma	752	1880	1375	485,576	4302	60,483	2610	487	472	18,090	17,253
TCGA-KIRC	Kidney Clear Cell Carcinoma	979	1880	1333	485,576	4399	60,483	2274	487	472	18,090	17,253
TCGA-BRCA	Breast	1260	1880	1418	485,576	4195	60,483	3671	487	472	18,090	17,253

## Data Availability

The molecular data, overall survival information, and other phenotypes from the TCGA and corresponding labels are available from NIH Genomic Data Commons (https://portal.gdc.cancer.gov/, accessed on 22 July 2025 ). The gene expression, miRNA expression, and DNA Methylation data was obtained from UCSC XENA (https://xena.ucsc.edu/, accessed on 22 July 2025). The CPTAC-LSCC and Moffitt LSCC data are available at [14,15]. The codebase for the project is available at https://github.com/lab-rasool/SeNMo accessed on 22 July 2025. The trained model checkpoints for the baseline SeNMo model along with the sample embeddings for future use are available in two parts at https://doi.org/10.5281/zenodo.14219799 (accessed on 22 July 2025) and https://doi.org/10.5281/zenodo.14286190 (accessed on 22 July 2025).

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
