# Peer review of "Self-Normalizing Multi-Omics Neural Network for Pan-Cancer Prognostication"

_ijms, 2025, doi:10.3390/ijms26157358_

Round 1

Reviewer 1 Report

Comments and Suggestions for Authors

The author has used the previously developed Multimodal Integration of Oncology Data System (MINDS) metadata framework, published in Sensors, 2024, to establish patient cohorts. The author has chosen 6 modalities, but this can be extended to other modalities, such as somatic mutations, etc. I am still missing the information on why only 6 modalities were chosen. Pre-processing steps seem to be convincing.

The Concordance index (C-index) of overall survival, 0.76, seems reliable, though achieving 1.0 is perfect. The Kaplan-Meier curves show that the survival for the low, intermediate, and high-risk groups diverges significantly.

SeNMo TLS ratio predictions show significant separation between high and low groups. Overall, SeNMo showed some reduced accuracy in a few cancer sets and limited validation.

Although TLS labels were image-derived and not directly integrated with pathologists, this approach to spatial transcriptomics, which maps gene expression to tissue samples, holds promise.

Finally, the author introduced SeNMo, a deep learning model for multi-omics data analysis, and making this approach publicly available can lead to further research of multiomics projects to facilitate cancer research.

Reviewer 2 Report

Comments and Suggestions for Authors

The authors of the paper presented SeNMo, a self-normalizing deep neural network trained on five heterogeneous omics layers, gene expression, DNA methylation, miRNA abundance, somatic mutations, protein expression, along with the clinical variables, and learns a unified representation robust to missing modalities. SeNMo can provide a baseline that can be readily fine-tuned 11 for diverse downstream tasks.

  • In section 2.3 the authors utilized different preprocessing steps. There are important preprocessing techniques that are normally used with this kind of data such as limmaand  DESeq2 for normalization or differential analysis. Were alternative methods chosen for specific advantages in multi-omics integration, and how do they compare in robustness to these widely adopted approaches?
  • In removing the law expression genes, how did the authors determine the threshold of 7. It is choice appears arbitrary without referencing. In common practice, median expression thresholds are used
  • I n features integration, how modalities were integrated early/late fusion, concatenation should be clarified.
  • Were the dimensionality reduction steps standardized for all multi-omics types, and if not, what criteria guided the modality-specific preprocessing?
  • Clarify the definitions of 'in-distribution' and 'out-of-distribution' for both data and tasks in Figure 1E.
  • It is not clear how the authors integrate clinical data with multi omics data? No mention for that in section 2.4.
  • The authors need to provide a figure to depict the architectural design of the SeNMo model,

Reviewer 3 Report

Comments and Suggestions for Authors

This paper presents SeNMo, a self-normalizing deep neural network designed for pan-cancer multi-omics integration and prognostication. Trained on >10,000 patients across 33 TCGA cancer types, SeNMo processes six modalities (gene expression, DNA methylation, miRNA, mutations, protein expression, and clinical data) to: Predict overall survival (OS), Classify primary cancer type, and Estimate tertiary lymphoid structure (TLS) ratios. The model uses self-normalizing layers (SELU + Alpha-Dropout) to handle missing modalities and demonstrates transfer learning capabilities with minimal fine-tuning. The authors provide open-access embeddings for downstream use.

1# The handling of missing modalities using zero-imputation is overly simplistic and may introduce sparsity bias. The authors should compare this approach to more advanced imputation techniques, such as variational auto-encoders.

2# The authors may like to provide an ablation study comparing SNNs to other architectures like transformers or CNNs to demonstrate their superiority.

3# The feature preprocessing step, which includes aggressive filtering (e.g., retaining gene expression levels >7 log2 FPKM), would definately discard low-expression genes that are still prognostically relevant. Authors must give persuasive reasons or rationale for this cut-off standard.

4# External validation is limited to only lung squamous cell carcinoma (LSCC) cohorts. The authors should validate their model on more diverse cohorts, such as those from the METABRIC or ICGC datasets, to ensure robust generalizability.

5# The sample sizes for the Moffitt-LSCC (n=108) and CPTAC-LSCC (n=77) cohorts are small, which may limit the robustness of the findings. Larger cohorts are necessary for more reliable validation.

6#  The failure of the model to achieve a reasonable C-index for testicular germ cell tumors (TGCT, C-index = 0.123) is concerning and remains unexplained. The authors should conduct subgroup analyses for rare cancers to understand these discrepancies.

7# The authors do not provide calibration plots (e.g., Brier scores) to assess the reliability of the predicted survival probabilities. This is crucial for evaluating the model's performance in clinical settings.

8# The prediction of tertiary lymphoid structure (TLS) ratios relies on manual segmentation from H&E/CD20 images, which is subjective and lacks inter-rater reliability. The authors should consider incorporating spatial transcriptomics or immune cell deconvolution to validate the TLS-associated gene signatures.

9# The 99.8% accuracy in primary cancer type classification is likely overfitted due to the curated nature of TCGA labels. The authors are required to validate this performance on external datasets, such as GTEx, to ensure generalizability. 

10# While the authors provide code and embeddings via GitHub and Zenodo, they do not offer a Docker or Singularity container, which is essential for ensuring reproducibility.

11# The model's black-box nature limits its clinical interpretability. The authors may try to include SHAP or Integrated Gradients (IG) analysis to identify the biomarkers driving the predictions.

Round 2

Reviewer 2 Report

Comments and Suggestions for Authors

After reviewing the authors' responses and the updated manuscript, I confirm that the authors have adequately addressed my comments and concerns. The revisions have improved the clarity and scientific rigor of the work, and I believe the manuscript is now suitable for publication in IJMS.

Reviewer 3 Report

Comments and Suggestions for Authors

All the comments are addressed.